# Thinking beyond Chatbots' Threat to Education: Visualizations to Elucidate the Writing or Coding Process

Badri Adhikari 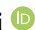

Department of Computer Science, University of Missouri-St. Louis, St. Louis, MO 63121, USA; adhikarib@umsl.edu

**Abstract:** Despite overwhelming evidence to the contrary, educational practices continue to be predominantly centered around outcome-oriented approaches. These practices are now thoroughly disrupted by the recent accessibility of online resources and chatbots. Among the most affected subjects are writing and computer programming. As educators transform their teaching practices to account for this disruption, it is important to note that writing and computer programming play a critical role in the development of logical and computational thinking. For instance, what and how we write shapes our thinking and sets us on the path of self-directed learning. Likewise, computer programming plays a similar role in the development of computational thinking. While most educators understand that "process" and "outcome" are both crucial and inseparable, providing constructive feedback on a learner's formative process is challenging in most educational settings. To address this long-standing issue in education, this work presents Process Visualizations, a new set of interactive data visualizations that summarize the inherent and taught capabilities of a learner's writing or programming process. These visualizations provide insightful, empowering, and personalized process-oriented feedback to learners and help to improve cognitive and metacognitive skills. Likewise, they assist educators in enhancing their effectiveness in the process-aware teaching of writing or computer programming. The toolbox for generating the visualizations, named Process Feedback, is ready to be tested by educators and learners and is publicly available as a website.

**Keywords:** Process Visualizations; process-oriented learning; self-directed learning; metacognition; process feedback

## 1. Theoretical Framework

### 1.1. Significance of Writing, Coding, and Metacognition

Writing shapes our thinking, regardless of whether we are writing to learn or learning to write [1,2]. Writing facilitates a logical and linear presentation of ideas that allows writers to explain their points of view. Similarly, learners write to clarify their thinking [3]. As presented in Emig's foundational work, writing is a process rather than a product [4]. Hence, the effective teaching of writing is an essential component of any successful school program. Such writing skills are best taught when related to specific and relevant content. This requires a more analytical approach to writing but provides a context for thinking deeply.

Just as writers craft sentences, computer programmers skillfully construct lines of code to create software that transforms our interactions with technology. Hence, in addition to writing, another cognitive skill taught widely in most schools and universities worldwide is computer programming. Several studies have pointed out that learning to code can improve various skills. Regardless of a student's major, learning to code exposes students' computational thinking [5]. A recent review on learning computer programming finds that learning to code can improve a learner's skills in mathematics, problem solving, critical thinking, socialization, self-management, and academic strategizing [6]. These findings suggest that the influence of learning to write a computer program on a learner's cognitive abilities can be profound. It could be far more than programming, leading to acquiring

powerful, general higher cognitive skills such as planning, evaluating, and monitoring abilities [7]. Learning to build useful software can serve as a means for learners to attain and improve widely applicable cognitive skills.

The recent accessibility of online resources reduces the motivation to write or code for those who let the chatbots write or code for them. While many learners are unfamiliar with the use of chatbots responsibly, those who primarily rely on them risk becoming overly dependent on such technologies. For instance, using resources to generate computer programs could particularly affect novice programmers by making them over-rely on such generative technologies [8]. Nonetheless, the role of writing in the development of logical thinking and the role of programming in the development of computational thinking is as important now as it has ever been.

While writing and coding develop logical and computational thinking, these cognitive skills alone are not sufficient to solve problems in authentic situations. Learners should develop *meta*-cognitive abilities to manage and regulate their problem-solving processes. Metacognition is commonly defined as the ability to think about one's own thinking. The text that inspired this definition was provided by John Flavell in 1979. He introduced it as the learner's "knowledge of their own cognition" and formally defined metacognition as the "knowledge and cognition about cognitive phenomena" [9,10]. Using metacognitive thinking and strategies enables learners to become flexible, creative, and self-directed. Metacognition also assists learners with additional educational needs in understanding learning tasks, self-organizing, and regulating their own learning. The National Research Council [11], stated, "Metacognition also includes self-regulation—the ability to orchestrate one's learning: to plan, monitor success, and correct errors when appropriate—all necessary for effective intentional learning... Metacognition also refers to the ability to reflect on one's own performance.".

The most intriguing and difficult part of teaching metacognition is that thinking is idiosyncratic. Brame (2019) [12], however, tells us not to be afraid to ask questions and to work on our own self-directed, self-monitoring, and self-evaluating plans. Effective self-directed planning involves understanding the overall goal and timeline of a project at hand. Similarly, self-monitoring involves investigating why one failed to master a needed skill, and self-evaluating involves assessing why one was or was not successful and efficient. Overall, learners can develop metacognitive skills when they ask themselves the questions needed to help them to "plan, monitor, and evaluate" their learning experiences.

While cognitive tasks such as writing and coding are learners' vehicles to organize their thoughts and develop higher-order thinking skills, metacognition does not develop on its own for all learners. Thus, it must be a target of learning supported explicitly in the classroom [13]. Langer (1986) [14] found that students at all grade levels were deficient in such higher-order thinking skills. Similarly, students are deficient in metacognition in self-regulated contexts also, such as when they are studying or practicing on their own [15]. Furthermore, today's outcome-oriented learning and education priorities make it more challenging to develop such skills. In general, developing metacognition and other related skills, such as self-awareness, understanding our own mind, and critical thinking, requires practicing reflection and mindfulness, learning from experience, engaging in self-experimentation, and seeking feedback. A potential solution to improving metacognition in learners is process-focused thinking aimed at developing process-oriented metacognition.

### 1.2. Importance of Process-Oriented Feedback on Learning

Research on learning, directly or indirectly, embraces the "process pedagogy". For instance, in direct opposition to the focus on written products, the 1970s and 1980s observed a groundswell of support for "process" approaches to teaching writing [1]. In writing, learners and educators use words such as drafting, prewriting, and revising in commonplace speech [3]. Similarly, in learning computer programming, educators often require learners to submit process data such as code comments, challenges faced, and code optimizations as a part of the assignment requirements. Some programming approaches, such as "test-first"

encourage programmers to write functional tests before the corresponding implementation code can be entered [16]. If process-oriented teaching and learning are so essential, why are today's education systems mostly outcome-oriented? One key reason is that it is not straightforward to assess a learner's process and provide feedback.

Feedback is one of the most powerful influences on learning and achievement [17]. If effective, feedback can promote student investment and independence in writing and coding [3]. For any feedback to be effective, several factors play a role. Hattie and Timperley [17] outline that feedback can be provided at four levels: task, process, self-regulation, and self. For instance, feedback provided at a task level may not generalize to other tasks but process-level feedback can be more effective in enhancing deeper learning. Another factor is the source of feedback. For instance, self-assessment can have a powerful impact on learner motivation and achievement [18]. Self-assessment is much more than simply checking answers. It is a process in which learners monitor and evaluate the nature of their thinking to identify strategies that improve understanding [19]. Similarly, peer feedback can also be effective. For example, a study to examine the effects of peer-assessment skill training on learners' writing performance found that learners who received in-depth peer assessment outperformed those who did not [20]. In addition, Narciss [21] describes how designing and implementing feedback strategies is complex and why a multidimensional model to design and evaluate feedback strategies is required. Such a multidimensional feedback model should consider several dimensions, such as the timing and function of feedback, adaptiveness, the learner's motivation, how feedback is delivered, task requirements, and the solution process.

Feedback on the process, in particular, can be considerably more powerful. Providing feedback on a learner's process can have a more positive impact on learning than feedback focused on the final outcome [22]. For example, process feedback can include a self-evaluation of a student's effort level or a formative assessment of progress regarding a specific goal [18]. Across a wide range of educational settings, formative-type assessments and feedback effectively promote student learning when they move beyond basic feedback that identifies whether a response is correct or not and provide elaborate feedback on a learner's thought process by explaining why it is incorrect or how to reach a correct outcome [22,23]. For learning, students perceive such elaborate feedback (EF) and knowledge of the correct response (KCR) to be more useful than only the knowledge of results (KR) and they show a positive attitude towards such feedback [24]. Formative assessments, in general, are a tenet of good teaching. Unfortunately, approaches to assessment have remained inappropriately focused on testing [25]. Pressures on education are threatening the use of formative assessments [26]. Teachers are, and continue to be, under enormous pressure to get their learners to achieve [27]. Moreover, in practice, these formative assessments make learners follow a prescribed and explicit process with *a lot of steps*, potentially developing *learned dependence* [28]. In addition, the practice of *lock stepping* learners by forcing them to divide their effort into stages conflicts with the recursive nature of the creative endeavor processes [3]. Hence, while feedback on the process is understood to be effective and valuable, in everyday educational settings, several challenges make the practice ineffective.

### 1.3. Computational Technologies to Cultivate Feedback on Process

In practice, while learners can be provided with feedback on their processes in several areas of the creative process, most computational tools primarily facilitate producing a better product. Process activities in writing are often subdivided into stages, such as prewriting, drafting, revising, and editing. These processes are recursive rather than linear and complex rather than simple [1]. To help learners to *prepare to write* or *prepare to code*, educators can offer learners models of finished writing or code, or, more effectively, can provide models on the spot, in front of learners, for the task in question. Models that demonstrate *how* to plan and write are more effective than the ones that demonstrate *what* to write [3]. In other words, in addition to providing professional examples, locally

produced examples and examples written by other learners of a similar age and skill level help learners to imagine how to craft a product incorporating similar features [3]. In this context, data visualizations of a learner's process can serve as visual cues to acquire feedback. For instance, the use of an effort visualization tool is found to increase team effort and improve performance [29]. In addition, with the help of visual analysis such as graphs and change indicators, feedback can become explicit and more effective [30]. In general, however, effective computational tools to explore, analyze, and learn from the process that leads to the product are missing. The crux of the problem lies in the absence of effective computational technologies for today's educators to demonstrate and discuss their own process of generating a product, or to present and analyze a *model* process, and for today's learners to acquire feedback on their process.

This article presents, discusses, and demonstrates how maintaining a self-monitored revision history (*snapshots*) of one's own text as a writer or a programmer's code at frequent intervals during the writing or coding process can be used to build interactive data visualizations that display and summarize the process. These Process Visualizations (PVs) can provide insights into the steps that a writer or a programmer took during their writing or coding process. This ongoing self-review by the author can facilitate effective self-feedback, peer feedback, and educator feedback. The toolbox developed, Process Feedback, is available at www.processfeedback.org (accessed on 1 June 2023).

## 2. Methodology and Implementation

This section describes the overall approach of retaining a user's process data (typed text of a writer or code of a programmer) and using these data to render informative data visualizations. First, the learner must realize that the web application being used is designed to allow users to write text or code. Once a user starts to type, the web application stores the text/code typed every five seconds in the user's local storage (Internet browser storage). These timestamped data contain the revision history of the user-typed text over the entire typing period. In other words, every entry recorded in this revision history contains both a timestamp and the text that the user was working with during the time point. Together, the full revision data collection tool captures how the typed text/code evolves (or changes) over time.

Transforming these typed text data (e.g., an English essay) with revision history information (i.e., time and full text at that time) into a format that can be used to develop visualizations requires that each text passage at any given time point can be connected to the text in the following time point. In this case, a passage is defined as either a sentence or a paragraph. These passage-to-passage connections between two versions of the text can then be used to observe the writer's creative flow when writing a passage, i.e., how a passage evolves (or changes) over time. For instance, as a writer adds content to a paragraph, the content of that paragraph will change with each time point. However, the various versions of a paragraph must be identified as the same to correctly display the paragraph's origin (i.e., the time point when it was introduced). An algorithm that usefully maps passages in a text at time $t$ with passages in the next step (at time $t + 1$) should also account for the fact that new passages can be added anywhere in the middle of the text at any time during the writing process.

Since identifying the same passages over multiple text revisions is a crucial step toward generating meaningful visualizations, we propose an algorithm for the same. The idea is to initially assign a random identity (ID) to each passage in each text revision, i.e., all passages are assigned different IDs regardless of their similarity. Next, these passage IDs are updated by calculating their similarities. Specifically, the update process starts from the most recent pair of two text versions. First, similarities between all passages at the time point $t + 1$ and all those at time $t$ are calculated (i.e., pairwise matched). For the two passages $p$ and $q$, which have the highest similarity score, where $p$ and $q$ are passages at time $t + 1$ and $t$, respectively, the ID of passage $p$ is updated to that of $q$ if the similarity is greater than a predefined similarity threshold. In other words, passage IDs are propagated

from the most recent time point to previous time points. This process is repeated for all adjacent time point pairs all the way to the first pair of revisions (see Algorithm 1).

---

**Algorithm 1** Identify same passages (paragraphs or sentences) between two versions of text

---

**Inputs:**

    *S*, list of text data at various time points; each item consists of an ordered sequence of passages with IDs

    *threshold*, similarity threshold (between 0 and 1)

**Outputs:**

    $S'$, IDs of passages at time $t_n$ replaced with matching passages at time $t_{n+1}$

 1: **procedure** UPDATEIDS

 2:     **for each** pair of time points $t$ and $t + 1$, starting from the latest pair **do**

 3:         **for each** $m \in S_t$ **do**                   ▷ $m$ is a passage at time point $t$

 4:             $n_{max} \leftarrow -1$       ▷ Stores the similarity of the most similar passage in $t + 1$

 5:             $n_{id} \leftarrow null$            ▷ Stores the ID of the corresponding passage

 6:             **for each** $n \in S_{t+1}$ **do**           ▷ $n$ is a passage at time point $t + 1$

 7:                 **if** SIMILARITY$(m, n) > n_{max}$ **then**

 8:                     $n_{max} \leftarrow$ SIMILARITY$(m, n)$

 9:                     $n_{id} \leftarrow n'$s ID

10:                 **end if**

11:             **end for**

12:             **if** $n_{max} > threshold$ **then**

13:                 $m_{id} \leftarrow n_{id}$       ▷ Update $m$'s ID so it is now identified as the same passage

14:             **end if**

15:         **end for**

16:     **end for**

17: **end procedure**

---

In the algorithm that identifies the same passages in two different time points, the similarity between any pair of passages is calculated using the normalized dot product of the *n*-grams of the two passages. Such a purely statistical similarity calculation based on *n*-grams is language-independent and works well when the lengths of the two passages are arbitrary [31]. Character-level n-grams are obtained with the *n* set to 5. Mathematically, if *p* and *q* are two passages being compared, where $p_i$ is an *n*-gram in *p* and $q_i$ is an *n*-gram in *q*, the first step is to calculate the weight of each distinct *n*-gram. In a passage *m* (either *p* or *q*), if $m_i$ is the number of occurrences of a distinct *n*-gram *i* among *J* distinct n-grams, the weight of *i* can be calculated as

$$x_i = \frac{m_i}{\sum_{j=1}^{J} m_j}, \tag{1}$$

where

$$\sum_{j=1}^{J} x_j = 1. \tag{2}$$

Then, for the two passages *p* and *q*, if $x_{pj}$ is the relative frequency with which the *n*-gram *j* occurs in passage *p* and $x_{qj}$ is the relative frequency with which the *n*-gram *j* occurs in passage *q* (out of all possible *J* n-grams in *p* and *q*), the normalized dot product can be calculated as

$$SIMILARITY(p, q) = \frac{\sum_{j=1}^{J} x_{pj} x_{qj}}{\sqrt{\sum_{j=1}^{J} x_{pj}^2 \sum_{j=1}^{J} x_{qj}^2}}, \tag{3}$$

Different from writing data, transforming the revision history data of code data (e.g., a Python program) into a format that can be used to develop visualizations has a different set of challenges. Unlike natural language text, code lines can have high character similarities and yet have considerably different meanings. For instance, in code, the only difference

between an inner for loop line and an outer for loop line serving different purposes can be two characters, *i* and *j*. Because of this small coding difference, performing an all-to-all line comparison between two versions of code is ineffective. Consequently, the approach that was used to split natural language text paragraphs into sentences cannot be applied to split a code into lines. Instead, given two versions of the unsplit code, a "difference" calculating algorithm is applied to find common subsequences or the shortest edits [32] on the entire code block. The central idea in calculating this "difference" is to formulate this problem of finding the longest common subsequence (LCS) and shortest edit script (SES) (known as the LCS/SES problem) as an instance of a single-source shortest path problem. The "npm diff" library implementation is used to obtain these code segments, each with a specific label, "common", "added", or "removed". These differences are obtained by considering each line as a unit, instead of considering each character or word as a unit. Following the idea similar to the one applied to natural language text versions, to obtain connections between the lines of code into adjacent versions of code, we initially generate random identities for all lines in all versions of the codes and update the identities of the sentences traversing backward from the most recent version of the code.

Regardless of whether the data for visualization are obtained from writing or coding, the visualizations rendered are dependent on the parameters supplied to the algorithm. Hence, interactive visualizations need to be rendered or re-rendered based on user-supplied values of the parameters. For instance, a user may provide a semicolon as an additional character to separate sentences. Similarly, another parameter is the *n*-gram size to calculate the passage-to-passage similarity and the similarity threshold to determine the identities of passages (or lines in the case of code). This approach of rendering visualizations based on user-supplied parameters makes the tool highly accessible across a wide variety of languages. For instance, not all natural languages have a period as the sentence separating character, and users should have the option to provide a custom sentence delimiting character(s) that applies to the language of the text being processed.

The single-page React web application (https://react.dev/, accessed on 1 June 2023) was developed using several recent technologies. Tailwind (https://tailwindcss.com/, accessed on 1 June 2023) and JavaScript were used to build the website. The site includes CKEditor (https://ckeditor.com/, accessed on 1 June 2023) to allow users to type text, and Judge0 (https://judge0.com/, accessed on 1 June 2023) [33], an open-source online code execution system, to allow users to code; the Monaco text editor and diff viewer (https://microsoft.github.io/monaco-editor/, accessed on 1 June 2023) to type and display code; and Plotly (https://github.com/plotly/react-plotly.js/, accessed on 1 June 2023) to display visualizations. The web application is hosted in Cloudflare (https://www.cloudflare.com/, accessed on 1 June 2023). A user's process data can either be downloaded and stored for the user's record only or saved to Cloudflare's R2 buckets. These data are encrypted using the Advanced Encryption Standard (AES) implemented in the 'npm crypto-js' library (https://www.npmjs.com/package/crypto-js, accessed on 1 June 2023), with a user-supplied passcode, and can only be accessed with the passcode to decrypt them. This effectively serves as authentication in accessing the data. Data saved on the cloud R2 buckets are additionally encrypted using Cloudflare's encryption-at-rest (EAR) and encryption-in-transit (EIT) technologies.

## 3. Results

The revision history data collected from a user can be cleaned, processed, and summarized to generate informative summary tables and interactive visualizations. Descriptive statistics, such as the total number of typed characters, words, sentences, lines, and paragraphs; the total and active time spent typing; and the average typing speed, can be presented as summary sentences or a table (see Figure 1). These quantitative summaries can be accompanied by data graphics displaying the actual data points. Similarly, the visualizations of the process data can facilitate drilling into the revisions; they also provide

an in-depth analysis of the overall process followed by a writer or a programmer. The next section discusses several interactive and non-interactive Process Visualizations (PVs).

**PV1: Process playback** (Figure 2). A highly accessible and easy-to-understand technique of visualizing a learner's process is to play back their typing. A short (e.g., 30 s) playback can display the typing steps that a writer or a programmer took. Options to pause, stop, and replay at a user-controlled speed can make such a visualization interactive. During playback, at each time point, trails of text deleted (if any) and the trails of text added can be highlighted so a viewer can track where the learner was typing at the time point. Such a process playback visualization may not provide any quantitative information but can be an engaging initial step toward further analysis.

| Summary and descriptive statistics | |
|---|---|
| Characters: | 2022 |
| Words: | 358 |
| Sentences: | 27 |
| Paragraphs: | 4 |
| Total Time Spent: | 29 minutes 53 seconds |
| Active Typing Time: | 16 minutes 26 seconds |
| Average Typing Speed: | 34 words/minute, 182 characters/minute |
| Time points: | 149 |

**Figure 1.** An example table of descriptive statistics.

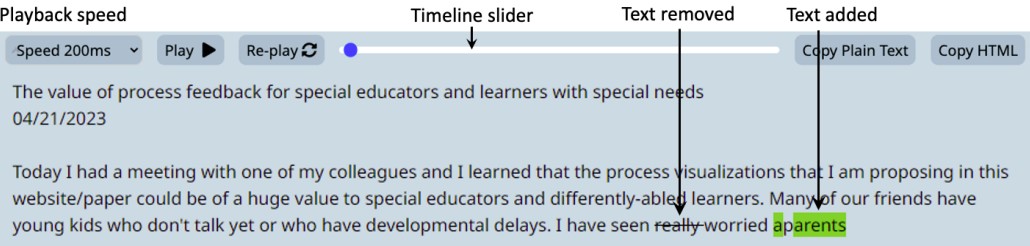

**Figure 2.** An example interface for playing back a writer's writing process. The green-highlighted copy shows text added at the time point and strikethrough formatting corresponds to text removed. The slider allows a user to control and view at any pace.

**PV2: Paragraph/sentence/line changes over time** (Figure 3). When analyzing one's writing or a programming process, it can be of interest to learn when a particular passage, paragraph, or sentence, in the case of writing and line in the case of code, originated or was removed. For instance, a sentence written at the beginning of the first paragraph may be pushed down throughout most of the writing time, only to later expand as a full paragraph. To visualize such process details at a passage level of interest, *stacked area plots* may be used with unique colors assigned to each passage [34]. Such an area plot can also show the relative size of the sentences or paragraphs and how they change over time. Additionally, these stacked area plots can be interactive, i.e., a viewer can see the actual text of the passage at any given time point by hovering over the stacks.

**PV3: Highlight active paragraph/sentence/line at each time point** (Figure 4). An interactive stacked bar diagram can display each passage as a stack with time marked by the x-axis. Such a detailed visualization can pinpoint (highlight) which passage a learner was working on at a specific time point during the writing or coding process. This visualization can also show how often a learner went back and revised a previously typed text. This can be achieved by highlighting only the passage being edited at that time point. An example application occurs when a learner is working on a revision task (starting with a bulk of text). In this case, the learner is looking at a stacked bar diagram to see if some paragraphs remain untouched.

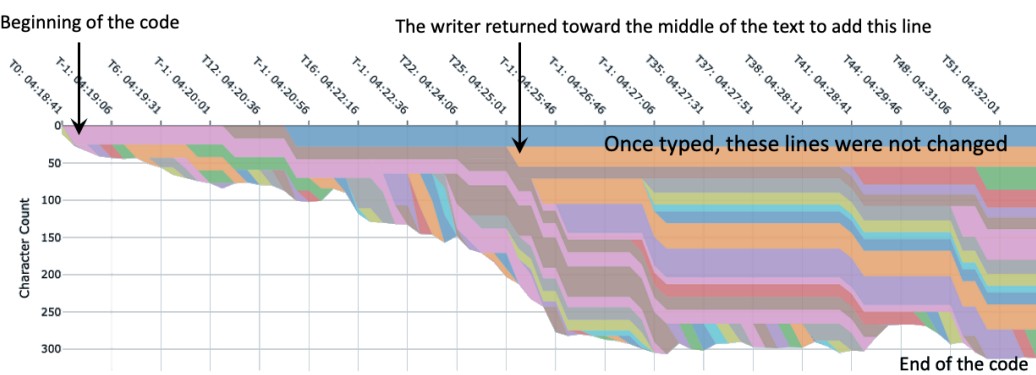

**Figure 3.** An example stacked area plot displaying line changes over time. This graph displays at what time point all the lines originated.

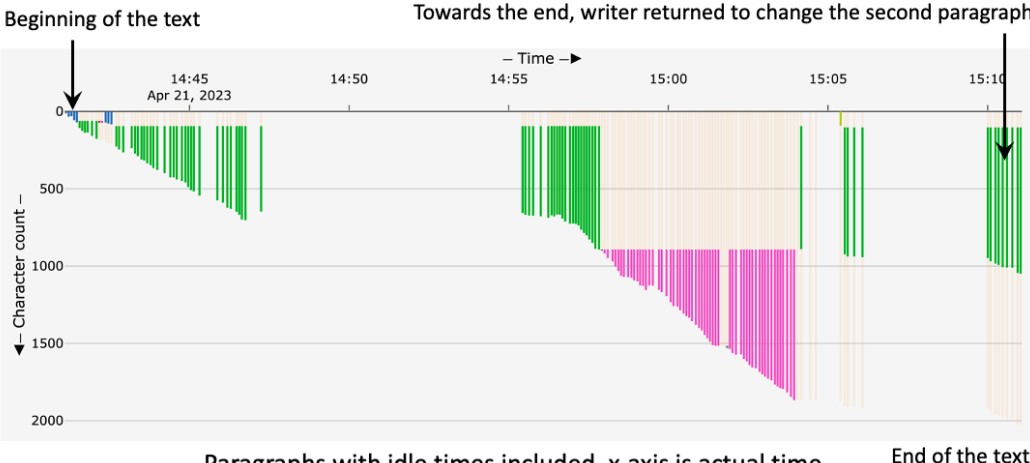

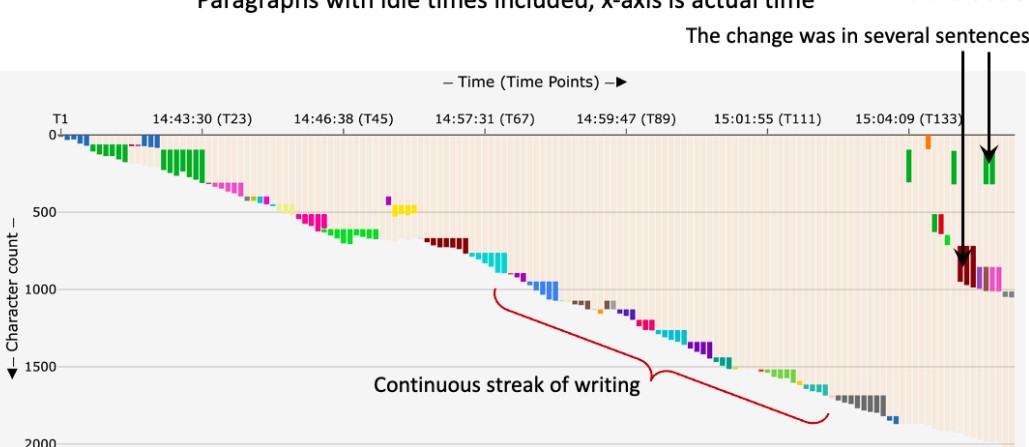

**Figure 4.** Example stacked bar diagrams highlighting active paragraph(s) (top figure) and active sentence(s) (bottom figure) at each time point. Each color corresponds to a paragraph in the top figure and a sentence in the bottom figure. Idle times are ignored in the timeline of the active sentence(s) plot. Hovering over any of the stacks displays the actual paragraph or sentence at the time point. Highlighted stacks represent the user's paragraph and sentence changes at the corresponding time point.

**PV4: Word frequencies** (Figure 5). In writing analysis, observing the frequencies of words typed by a learner can be informative. For instance, self-directed learners may be interested in observing if they overuse the article *the*, the adverbs *very* and *respectively* or conjunctive adverbs like *however* and *consequently*. Such observations may motivate

them to increase their vocabulary and check the accuracy and necessity of overused and misused words. Although word frequency data can be calculated from the final text independently of the process, the frequency of words removed can also be informative. One caveat of counting words only based on user-defined character(s) but without the knowledge of a language is that these counts can be inaccurate for languages such as English, where the same letters can be in lower or upper case. For instance, in such language-independent word frequency analysis, "the" and "The" would be identified as separate words. Word frequency can be plotted as standard bar diagrams displaying a certain percentage or a certain number of the most used words and their corresponding usage frequency throughout the document.

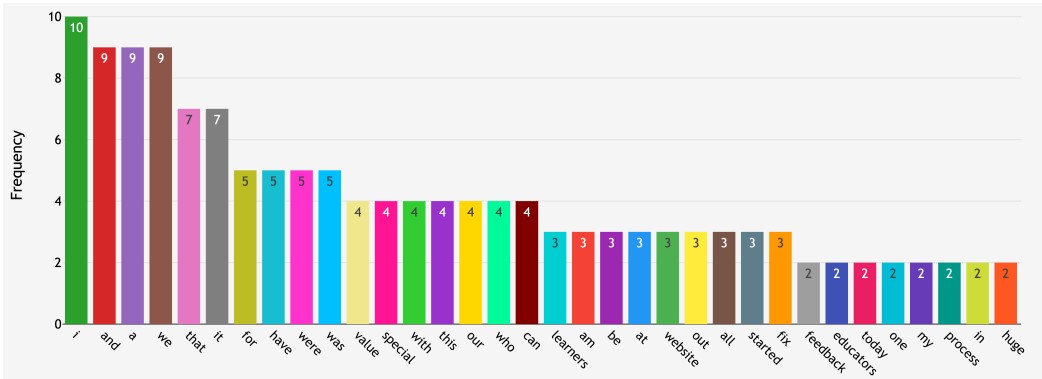

**Figure 5.** An example bar diagram displaying word frequencies.

**PV5: Passage-to-passage pairwise similarities** (Figure 6). Writers can also fall into the habit of repeatedly using the same phrases that fail to inform or can be deleted, such as *it should be noted that* or *it is well known that*. Visualizing pairwise similarity between all sentences in the final version of the text can help to identify such habits. A visualization that serves this purpose is an interactive heatmap where each cell represents a similarity score between pairs of sentences. Displaying the actual sentence pairs by allowing them to hover over the cells can add interactivity.

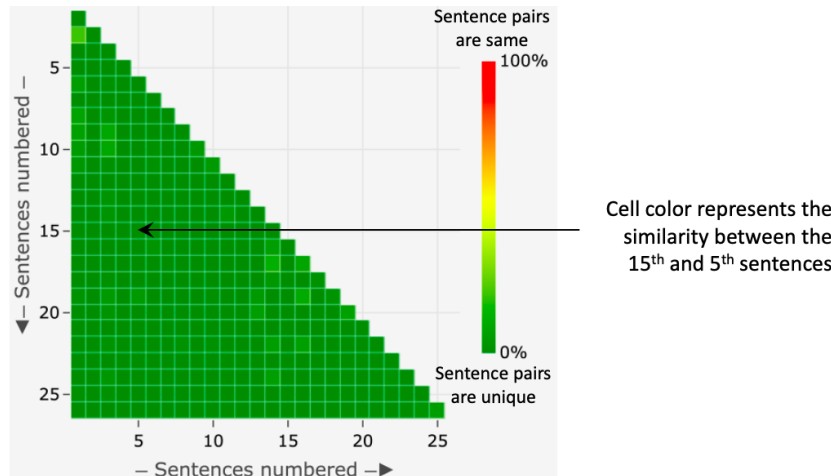

**Figure 6.** An example heatmap plot to display pairwise passage-to-passage similarity. Both x-axis and y-axis represent the sentences shown in the order in which they appear in the final writing. Hovering over any cell of the heatmap shows the two sentences and their similarity score.

**PV6: Words/characters change over time** (Figure 7). Changes in the number of words or characters over time can be visualized as a line diagram for cumulative and as a lollipop chart (or a bubble chart) for non-cumulative review. In a bubble chart, the bubble sizes can

represent the number of characters added or removed. One approach to display bubbles at a consistent location in the y-axis is to set the y-axis based on the size of the change. Since changes can have large variation, data can be log-normalized.

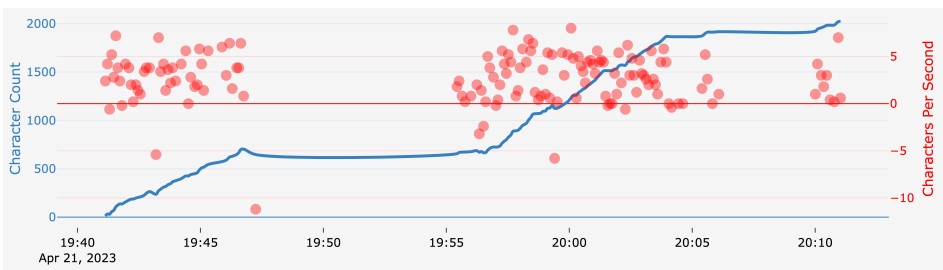

**Figure 7.** An example dual-axis scatter and line plot displaying an increase in the total characters typed over time points (blue line) and characters per minute at any time point (red points).

**PV7: Interactive any-to-any revision version comparator** (Figure 8). Users interested in an in-depth analysis will want to observe the text added or removed between any two time points in the revision history. Users also need an interactive visualization tool that enables them to browse the entire timeline and select a range to view the changes. To do so involves effectively selecting two time points. This can be achieved by combining three tools: (1) a date-based navigator, (2) a difference viewer (i.e., the Monaco diff viewer), and (3) a timeline chart (i.e., a bubble chart), which can show all the time points. Users can then interact through the bubble chart or navigate directly through the diff viewer. Such an interactive visualization can facilitate in-depth analysis, allowing the user to ask why a sentence or phrase was removed at a certain time.

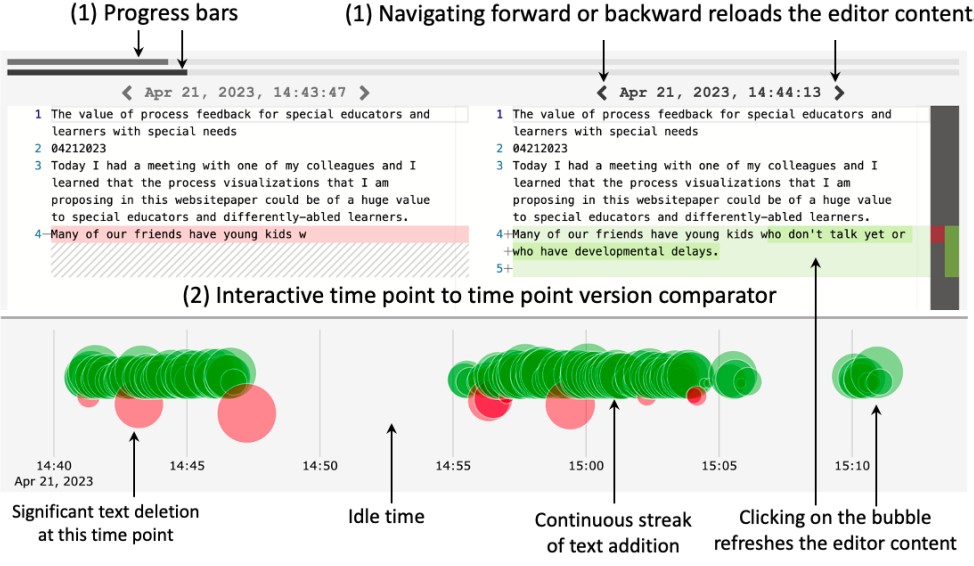

**Figure 8.** An example interactive any-to-any revision version comparator consisting of three components: (1) two date navigators at the top with arrows on either side of the dates, with progress bars on top connected to the dates (top progress bar for left date and bottom progress bar for right date); (2) a difference viewer shows two versions of text dictated either by the clicking on the date arrows or by clicking on the bubbles in the chart at the bottom; (3) a bubble chart shows the timeline, where the bubble sizes correspond to the number of characters added or removed. Panning and zooming on the bubble chart facilitate easy micro/macro analysis. Clicking on any bubble loads the changes made in that particular version in the difference viewer windows. Green bubbles indicate that text was added at that time point and red bubbles indicate that text was deleted. In addition, hovering over the bubbles shows the total number of characters added/removed.

**PV8: Timeline showing successful and unsuccessful code executions** (Figure 9). In the case of programming tasks, displaying a timeline plot with points indicating the executions of code and highlighting both successful and unsuccessful executions can help users to review how often they executed their code.

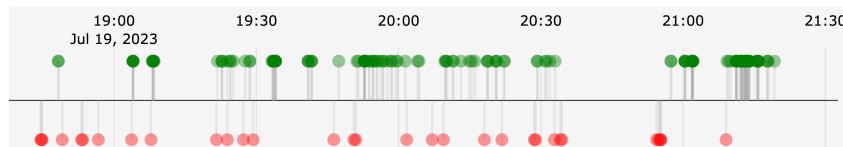

**Figure 9.** An example timeline showing successful executions (green points) and **executions with errors** (red points) in a computer programmer's coding process. Hovering over the points shows the actual output or error message.

## 4. Discussion

### 4.1. Exploratory Data Visualizations as Clarifying Tools

Data graphics are widely used as an explanation tool. In a more general, all-inclusive context, data graphics can serve the purpose of explanation, exploration, or both. While explanatory data graphics can communicate a point or display a pattern or a concept, exploratory data visualizations invite the viewer to discover information [35]. While discussing the fundamental principles of analytical design, in his book "Beautiful Evidence" [36], Edward Tufte lists "completely integrating words, numbers, images, and diagrams" as one of the core principles. Although these exploratory data visualizations can appear complex at first, they can be vehicles for new insights. Alberto Cairo, in Truthful Art [37], noted insightfulness as one of the most important qualities in his presentation of *five qualities* of great visualizations. Adding interactivity to visualizations can strengthen this integration. Interactive and integrative visualizations can facilitate two fundamental acts of science: description and comparison. Several aspects of interactive data visualization can, however, make them inaccessible. For increased public accessibility, these visualizations should be improved for user-friendliness and high usability.

Utilizing interactive visualizations to display a learner's process and provide feedback represents a pedagogical shift that exhibits several advantages over conventional teaching methodologies. These visualizations can be instrumental in illustrating the writing or coding process in real time, enabling immediate feedback, and increasing learner engagement. Observing and retaining the process further allows learners to evaluate their development or shifts in writing or coding habits over time. For such process-aware pedagogy to be successful, educators must acquire proficiency in interpreting the visualizations and engaging with them interactively. A lack of prior familiarity or formal training with these data visualization tools might engender initial resistance. Nevertheless, the use of Process Visualizations harbors substantial opportunities to enhance teaching and learning efficacy, and the resultant benefits surpass those attainable through traditional instruction without them.

### 4.2. Acquiring Feedback from Insightful Process Visualizations

The Process Visualizations clarify the concepts and steps of the process. This provides tools for the acquisition of feedback—from oneself, a peer, or an educator—on several dimensions of a writer's or programmer's process. For instance, if a writer is focused on improving their prewriting skills, these visualizations can enable the exploration of how frequently the writer switches to other modes, such as revising and editing (see Figure 4). While it may be recommended to finish prewriting and then focus on revision, it is possible to polish paragraphs while writing. Prewriting embodies the initial steps of writing. A writer's approach is usually an iterative process; thus, it can be insightful to learn what process a writer follows to finalize an initial prewrite. Similarly, in learning and practicing the "test first" approach to programming, Process Visualizations can enable the exploration of how often the programmer revises the test cases as they start to code the actual solution and how often they revise the solution as the test cases evolve. In general,

learners could also use these visualizations for self-revelation, direction, and improvement, and the visualizations can provide immediate answers to questions such as (1) where in the process they spent the most time, (2) how much time they spent creating the first draft vs. revising and editing the draft, (3) which paragraphs were edited and revised, and (4) which paragraphs were not edited. Overall, the visualizations facilitate exploring, looking back, and looking into one's process and approach. Furthermore, learners can engage in *inquiry* by focusing on the process, regardless of whether the processing was originated by the learner, a peer, or an educator. Exploring and understanding the theories behind teaching and learning, followed by the engagement of all parties toward making the most of every learning experience, is the portal to learning through communication, feedback, improvement, and success. Engaging in inquiry and agreeing on the *theories that drive teaching and learning* is powerful. Whitney [3] summarized theorizing and engagement as follows:

> "Engaging in inquiry means not only learning practices recommended by others or perfecting the practical execution of a set of teaching strategies but, rather, theorizing about teaching and learning in a way that then frames future interpretation and decision-making." (Beyond strategies: Teacher practice, writing process, and the influence of inquiry, 2008, p. 205).

*4.3. Securing Personal Process Data*

Individual creations such as a piece of writing, code, or art are intellectual properties, and their security is paramount in the contemporary digital landscape. The process data from an individual (i.e., the revision history) can be at least as important as the outcome. However, process data tend to be much more sensitive and personal than the final outcome. Such data could be tampered with or exchanged with malicious intent. For instance, if trained on process data, deep learning methods could potentially generate "deep process fakes", which can have disastrous and unintended consequences. The deep fake scenarios [38] have brought about the ability to alter photos and videos to the point of misrepresenting a person's political stance, corporate identity, or involvement in a crime. Hence, securing and sharing personal process data with only trusted parties is critical. It is also crucial to develop strict policies and restrictions for machine or deep learning algorithms that are applied to individual process data. The current version of the Process Visualization techniques proposed in this work, including the n-gram similarity calculations, does not use any form of machine learning or deep learning.

*4.4. Learners Are Not Monitored or Controlled during Their Creative Process*

The current release of our web application does not save any user data to our online servers, except at required times, such as the time of loading the single-page web application, downloading a task description (question), explicitly saving a question or a response to the online storage server, verifying a human user, or when executing a computer code (to compile). However, any third-party components used could be using their own data collection techniques. All revision histories of a user's data are saved locally in the user's Internet browser's local storage. In other words, users have complete control over the process data that they share and when. For instance, the web application does not track whether a learner working on a project clears their history, starts over, or abandons the writing process altogether.

The web application can be used entirely as a self-improvement tool. If learners wish to share their process, they can self-review it first and then decide if they want to share it. The current version of the application does not have any authentication built in (e.g., logging in or signing in). The intent here is to make the website highly accessible to users since requiring an email address and forgetting passwords will not be a concern. Hence, users are not required to include any private information. *If a user writes their content, views the Process Visualizations, and downloads the data without saving them to our servers, our application has no knowledge of any such process and does not attempt to track them.* In sum, users have a

clear choice to discontinue revision logging at any time, and if they do decide to share, they can review their process data first on their own.

If the technique of maintaining a revision history was to be used in non-transparent ways (i.e., in exams, tests, or standardized language/programming tests), learners could feel vulnerable, monitored, and even controlled. This type of non-transparency could cause learners to resist using any education/learning system if given a choice, including the system presented herein. As an alternative, in a setting where learners feel monitored or controlled, an educator can motivate their learners by showing (modeling) their creative process using visualizations. For instance, a writing instructor can demonstrate how common it is to delete text during a revision process and an educator teaching programming could demonstrate how common it is to run into errors and spend time debugging.

*4.5. Educators' Assessment Time*

One concern that may be raised when exploring and interpreting Process Visualizations is that educators' assessment time could increase. It can be argued that educators worldwide are already struggling and need more time to provide feedback that promotes learning. Moreover, educators are increasingly concerned about how to build students' self-efficacy [39]. Unfortunately, finding time to develop tools that can help to build students' self-efficacy and promote learning is a hurdle that not many educators can overcome. Teachers and students only have so many hours each day, which is usually strictly regimented. Additionally, teachers cannot possibly know all the individual problems that students have had in their learning process. At times, not even the student knows the problems that are causing barriers to their learning. How, then, will educators be able to evaluate the process of learning and provide feedback on the process? This is a genuine initial concern; however, several premises must be considered.

First, compared to reading an outcome (essay or code), visualizations can be faster to interpret, particularly when the structure of the visualization is consistent, with only the data changing. The first few exploratory activities and interactions will take time. However, once the structure and features of the visualizations are learned, the focus will naturally shift to seeing the story being unveiled by the data. These visualizations, however, should be improved for the highest accessibility and user-friendliness.

Second, after exploring a few varieties that may be possible within a visualization structure, users will develop knowledge about the range of possible variations and gradually become efficient at interpreting meanings from the visualizations. Short and dedicated tutorial videos on each visualization, aimed at enhancing visualization literacy, can also help the users to learn how to interact with the visualizations.

Third, not all learners and educators should use the visualization toolbox for every task. Learners may only use it occasionally to learn about their style or with their peers to learn from each other. Similarly, educators may use it only for a specific task or for a learner who would benefit from such feedback.

Fourth, in situations where the outcome is more important than the process, learners and educators can discard the Process Visualization approach altogether because these visualizations do not replace the outcome; they only supplement it. The premise of this research is that if learners and educators appropriately divide their time between reading the completed writing or coding tasks and exploring the Process Visualizations (i.e., consider both: the outcome and the process), the two can complement each other and together serve as a powerful resource to either self-learn or acquire feedback from others.

**5. Conclusions**

This article presented and described Process Visualizations, a new set of interactive data visualizations aimed at visualizing a learner's writing or programming process. These visualizations summarize a learner's style and habits and enable looking back into the process trail. They can also encourage writers and programmers to be process-aware and serve as tools to analyze their creative and cognitive processes, ultimately facilitating them

to acquire feedback on their processes. Consequently, this work provides a toolbox to elucidate a learner's creative process and addresses a long-standing challenge in education.

**Funding:** This research was supported in part by startup funds provided by the Department of Computer Science, University of Missouri-St. Louis.

**Institutional Review Board Statement:** Not applicable.

**Informed Consent Statement:** Not applicable.

**Data Availability Statement:** The Process Feedback website is ready to be tested by educators and learners and is available at https://www.processfeedback.org (accessed on 1 June 2023).

**Acknowledgments:** I am deeply grateful to the individuals who have contributed to the development of this manuscript. Their expertise and thoughtful insights have been essential in shaping and refining the content. Specifically, I extend my heartfelt appreciation to Shea Kerkhoff, Abderrahmen Mtibaa, Sambriddhi Mainali, Jill B. Delston, Jason Wagstaff, and graduate students Shaney Flores and Kate Arendes at the University of Missouri-St. Louis, as well as undergraduate student Nilima Kafle, for engaging in several stimulating discussions and for dedicating their time to providing insightful feedback on the manuscript. Furthermore, I would like to acknowledge Manu Bhandari and Khem Aryal at Arkansas State University, Jie Hou from Saint Louis University, as well as Bishal Shrestha, Milan Adhikari, and Nitesh Kafle from Kathmandu, Nepal, for their valuable discussions and comments on the manuscript. I am also grateful to Amber Burgett at the University of Missouri-St. Louis and Laura March at the University of Missouri-Columbia for suggesting several useful readings. Finally, I am grateful to Carla Roberts at Preferred Copy Editing for editing the manuscript at short notice.

**Conflicts of Interest:** The author declares no conflict of interest.

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
