# Peer review of "Thinking beyond Chatbots’ Threat to Education: Visualizations to Elucidate the Writing or Coding Process"

_education, doi:10.3390/educsci13090922_

Round 1

Reviewer 1 Report

General remarks:

The overall manuscript is relevant information for existing literature. I would suggest to look more into formal and academic language use (some sentences are more popular-scientific; e.g., see first sentence of the abstract). Due to the topic, this manuscript displays potential albeit there is quite some work needed to be of sufficient quality in terms of conceptualization and coherence.

Specific remarks:

p.1.2                What are large language models? Moreover, the sentence after that is also unclear. So you are talking about revisions or changes in language use in teaching and learning situations? If this is the case, please clarify the sentence.

p.1.5                “Writing and…” = Comes as a surprise (it does not follow logically after the previous sentence). Aim for more coherence. I would go over the complete abstract. Your reasoning doesn’t come across as logical nor coherent.

p.1.9/10           Do you imply that this is not common practice elsewhere? What is the take-home message from this statement?

p.1.12              The “(and learning)” is redundant.

p.1.15              I would avoid mentioning the location of the toolbox (thus place the website in a footnote).

p.1.16/17         Start the sentence differently to make it more logical.

p.1                   I think you are mixing the Introduction and Theoretical framework. If this is the case, please adjust the header (Introduction = Theoretical framework) unless the journal instructs you to use the “introduction” header.

p.1.23              “During the process” = During this process.

p.1.25              “a prominent…scholar” = redundant.

p.1.27/28         Explain what the contribution was rather than mention that the author contributed.

p.1.32/33         Relevant. This is such a crucial argument. I would suggest to mention this in the abstract as well. If readers only read the abstract, they want to read this crucial statement.

p.1                   The sub headers do not help to create coherence. You need to work with connecting sentences. Also make use of this for the other sub headers.

p.1.37              What is the relevance of a 1984 source?

p.2.53              I would suggest to mention another example of those higher cognitive skills. I would suggest monitoring. In addition, I would list a source.

p.2.56              Do you mean learning metacognition (as in: metacognitive skills can be trained/learned) or do you mean metacognition applied to learning situations? Furthermore, embed the quote into the main text. Why mention the book title?

p.2                   I think I spot a few double space before the start of a sentence. Also check the remainder of your manuscript for this.

p.2.78              This would be better introduced if you create more coherence in the previous sections.

p.2.83/84         You refer to the role of metacognition in the domain of mathematics; however, you refer to cognitive tasks in the sentence after that. This is confusing for the reader and, in particular, if the reader does not know how these concepts differ. The link to Langer (1986) is insufficient. I would also suggest to Zimmerman and other more recent works that addressed this topic.

p.3                   Header and sub header raise different expectations. Create more coherence and match the headers to that.

p.3.110            Do not start a sentence with “and”.

p.3.111–119    You provide a one-dimensional view of feedback. The success of feedback depends on several factors that you do not mention here (e.g., type of task, type or format of feedback). Link from instructor or computer to peer feedback is incoherent. For process-oriented feedback, Hattie and Timperley are researchers you want to refer to. In addition, Narciss also provided a detailed feedback model that will help you shape your theoretical framework.

p.3.132/133     I would not end a paragraph with this. I have questions that won’t be answered.

Discussion      You are aiming for visual data (i.e., you place an emphasis on visualization tools). This requires that you elaborate more on the role of visual cues in feedback. The introduction does not provide the reader with sufficient knowledge/information.

                        Why do you use a question as a header? This is a bit odd. Can you rephrase this as a statement?

See above. 

Reviewer 2 Report

·        The abstract is well-written overall, with a clear outline of the issue being addressed and the solution proposed. However, some parts can be further improved for clarity.

1.      I've noticed you've used the term Process Visualizations a lot but didn’t elaborate on it anywhere. I think it would be a good idea to give a very brief description of what it actually means for people who aren't familiar with the term either in the abstract or in the introduction section.

2.      Make the impact of your work clearer in your final sentence of the abstract, rather than just saying “will facilitate learners’ ability to acquire higher-order skills 17 such as self-directed learning and metacognition”.

·        The introduction is very well put together and is really simple and clear to understand. But there are a few areas that could be polished for improved clarity.

1.      When mentioning the impact of chatbots, clarify why it is a problem and why it should be addressed. The current text states, "However, the recent use of online resources and chatbots reduces the motivation to write for users who let the chatbot write for them." This could be expanded upon to detail why this is detrimental to the learning process.

2.      The part about coding needs a little more information to be on the same level as the part about writing in terms of clarity for the audience.

Remember that the introduction sets the tone for the whole paper and can do a lot to get people interested.

·        Your methodology section provides a detailed explanation of your process of data retention and its subsequent usage in rendering informative data visualizations.

1.      Personally, I feel the methodology section to be really good, but it might be a bit complex and readers might find it difficult to follow (Had that experience personally). So I would recommend maybe fine-tuning it and simplifying it a little. Remember that the methodology section should be detailed enough that another researcher could replicate your study, but also clear and straightforward enough that readers with varying levels of expertise can understand it.

·        Furthermore, the results sections and discussion have been really well put. Great work.

 Overall, the article is a clear and well-structured academic article, effectively presenting complex concepts and research findings in an accessible manner for readers across various disciplines.

Reviewer 3 Report

The paper presents a process-oriented teaching and learning approach. The paper also provides good contextualisation to the visualisation of the process data that can facilitate drilling into the revisions with an in-depth analysis of the overall process. It introduced several Process Visualisations to promote the understanding of what writers/programmers should perform to acquire feedback on their processes.

The paper provides a web tool ready to be tested and publicly available at www.processfeedback.org; this is an excellent instrument to complement the theoretical reference addressed in the paper. However, the Conclusion needs to be reviewed because it is too generic, does not highlight the key issues or main achievements addressed, should not include many broad examples about possible applications or target sectors, and should focus more on resuming what is relevant to retain from the process-oriented teaching approach.

It could be interesting to consider a SWOT analysis or at least compare the benefits of a process-oriented teaching approach with traditional teaching approaches. 

Round 2

Reviewer 1 Report

The first two sentences do you logically follow (lack of coherence). In a similar vein, the topic switch to "writing" comes as a surprise. Also the connection to computational thinking via programming does not follow logically. The last sentence of the abstract means that you wil discuss cognitive and metacognitive skills.

In your TK, the topic of computer programming comes as a surprise. Introduce this better. 

p.2.42 = double space before the start of the sentence?

I would also say that students do not know how to use these chatbots in a responsible way. The use of it per se is not a bad thing. The fact that they do not reflect upon certain aspects of it diminishes their (meta)cognitive development.

Link to cognition comes as a surprise. Overall, your efforts to increase coherence aren't paying off (albeit I do notice efforts in doing so by using signaling words).

The suggestions I gave for feedback are relevant but do not share the complete story. What are examples of feedback focused on the process? How is KR and KCR linked to this? You discuss the feedback without making it concrete. This becomes more apparent since you are talking about visualization options as feedback. 

Round 3

Reviewer 1 Report

The revision of the abstracts results in more questions. Writing and computer programming aren't necessarily part of teaching (this link has not been made explicit).

Do you refer to student needs, or from teachers? From management? Terminology is inconsistent. Compare outcome-driven with product-oriented. A few sentences later you use outcome-centric. In addition, it is unclear to what "educational needs" refers to (i.e., either outcome or process).

The two needs you describe do not follow logically after what you have written. 

There are a few extra spaces inserted before some of the new sections (e.g., p. 3.116). 

Author Response

Dear Esteemed Reviewer,

Thank you again for your comments on our revised submission. We apologize for making unrequested changes and creating confusion. We reverted the first half of the abstract to the previous version and addressed the previous round of comments instead of making unrequested changes. Below, we present a detailed point-by-point account of how we have addressed your comments. In the revised manuscript, the changed text is in red.

Point 1: "The first two sentences do you logically follow (lack of coherence). In a similar vein, the topic switch to "writing" comes as a surprise. Also the connection to computational thinking via programming does not follow logically. The last sentence of the abstract means that you wil discuss cognitive and metacognitive skills."

Response 1: To address this comment, here are the changes we have made in the abstract:

  1. The beginning of the second sentence in the abstract is revised so the first two sentences follow logically.
  2. To make the topic switch to writing not abrupt and create a context, we have a new sentence before the topic switch: “Among the most affected are teaching writing and computer programming.”
  3. To connect computational thinking to programming more logically, we have merged the two parallel ideas in a single sentence. The part of the sentence that achieves this is: “it is important to note that writing and computer programming play a critical role in the development of logical and computational thinking.”
  4. Also, as before, the last sentence is revised and inserted before the end of the paragraph. Since this change was not commented on, we understand that this comment was already addressed in the previous revision.
  5. In light of the new comments, we have replaced “outcome-driven” with “outcome-oriented” in the abstract so the terminology is consistent.

Point 2: There are a few extra spaces inserted before some of the new sections (e.g., p. 3.116).

Response 2: We checked this thoroughly and confirmed no double-spacing before any sections. We examined the issue of the “double space” appearance by doing some tests and found that the formatting is controlled by the Latex template provided by MDPI. In an attempt to not stretch the characters, it appears that Latex shows some extra space between sentences even when the space character is only one. In this case, if we remove the long word “self-evaluation” from the sentence, the spacing looks right.

Thank you for your review. We are grateful for your time and effort in reviewing our manuscript.